# Polysaccharides: The Sweet and Bitter Impacts on Cardiovascular Risk

**DOI:** 10.3390/polym17030405

**Published:** 2025-02-03

**Authors:** Grzegorz Kalisz, Joanna Popiolek-Kalisz

**Affiliations:** 1Department of Bioanalytics, Chair of Dietetics and Bioanalytics, Medical University of Lublin, Jaczewskiego 8b St., 20-090 Lublin, Poland; 2Department of Clinical Dietetics, Chair of Dietetics and Bioanalytics, Medical University of Lublin, Chodzki 7 St., 20-090 Lublin, Poland; 3Department of Cardiology, Cardinal Wyszynski Hospital in Lublin, al. Krasnicka 100, 20-718 Lublin, Poland

**Keywords:** dietary polysaccharides, fiber, cardiovascular risk, glucan, obesity, metabolic syndrome, cardiovascular prevention, dyslipidemia

## Abstract

Cardiovascular risk is a clinical factor that represents the probability of developing cardiovascular diseases (CVDs). This risk is shaped by non-modifiable and modifiable factors, including dietary patterns, which are the main lifestyle factor influencing CVD. Dietary polysaccharides, integral to nutrition, have varying effects on cardiovascular health depending on their type and source. They include starches, non-starch polysaccharides, and prebiotic fibers, categorized further into soluble and insoluble fibers. Soluble fibers, found in oats, legumes, and fruits, dissolve in water, forming gels that help lower serum cholesterol and modulate blood glucose levels. Insoluble fibers, present in whole grains and vegetables, aid in bowel regularity. The cardiovascular benefits of polysaccharides are linked to their ability to bind bile acids, reducing cholesterol levels, and the production of short-chain fatty acids by gut microbiota, which have anti-inflammatory properties. However, not all polysaccharides are beneficial; refined starches can lead to adverse metabolic effects, and chitosan to mixed effects on gut microbiota. This review examines the dualistic nature of polysaccharides, highlighting their beneficial roles in reducing cardiovascular risk factors and the potential adverse effects of specific types.

## 1. Introduction

The probability of an individual developing cardiovascular diseases (CVDs), such as coronary artery disease or stroke, is referred as cardiovascular risk. This risk is influenced by a variety of factors, which can be broadly categorized into non-modifiable and modifiable risk factors [1]. Part of the modifiable factors are dietary patterns, well-known among physicians, researchers, and patients, and considered very important in modern lifestyle medicine. To assess CVD risk in patients, a Framingham Risk Score and the American College of Cardiology/American Heart Association (ACC/AHA) risk calculator are used in the United States, while the Systematic Coronary Risk Estimation 2 (SCORE2) is the standard for European countries. All of them are based on recognized CVD risk factors such as systolic blood pressure, lipid profile, smoking status, and sex [1]. Elevated blood pressure and diabetes promote endothelium dysfunction and damage, which along with elevated low-density cholesterol (LDL) levels leads to atherosclerosis development. If atherosclerotic lesions are present in coronary arteries, such a condition is defined as coronary heart disease. CVD is a major cause of death worldwide [2]. This is why the impact of lifestyle aspects, including diet, on modifiable CVD risk factors such as hypertension and dyslipidemia is being investigated extensively [3,4,5].

Polysaccharides, carbohydrates consisting of monosaccharide units forming longer chains, are integral components of the diet. They play a crucial role in energy intake and other physiological functions [6]. Their role in health, particularly cardiovascular disorders, is an area of active research, revealing risk factors and beneficial effects, depending on the investigated compound. Dietary polysaccharides are usually categorized based on their sources and structures, and examples of their differed structures are presented in Figure 1. Most reported macromolecules in health-related studies have been derived as either starches, non-starch polysaccharides like dietary fibers, or specialized prebiotic fibers [7]. Fibers can be categorized into soluble (pectin, beta-glucan, guar gum, or mucilages) and insoluble fibers (cellulose, hemicellulose, lignin). Cellulose, hemicellulose, pectin, and beta-glucans are generally indigestible by humans, but play an important role in gut health and metabolic regulation [8,9]. Cellulose is widely found in broccoli, cabbage, kale, and green beans [10]. In fruits, apples, pears, and berries contain cellulose, as do whole grains, and legumes like beans, lentils, and peas [11,12]. Moreover, hemicellulose is abundant in whole grains, nuts, and seeds, including sunflower seeds, flaxseeds, almonds, and root vegetables like carrots, potatoes, and turnips. Legumes such as soybeans, beans, and chickpeas are also rich in hemicellulose [13]. Pectins are particularly commonly found in fruits, with high concentrations in apples, oranges, grapefruits, lemons, and other citrus fruits. Known for their gelling properties, they are widely used as a setting agent in jams and jellies. Similarly, beta-glucans present gel-like properties, but contrary to pectins, they are reported to be biodegradable in the digestive system. They are present in oats, including oatmeal, bran, and whole oat products, but also in barley grains, barley flakes, and barley flour. The most prominent sources of the differences between these molecules are in their molecular structures and viscosity. Oat beta-glucans typically consist of linear chains of β-(1→4) and β-(1→3) linked D-glucose molecules with a ratio of about 3:1. Similar to oat beta-glucans, barley beta-glucans also contain β-(1→4) and β-(1→3) linkages, but with a ratio closer to 2:1.

Certain types of mushrooms, such as shiitake, maitake, and reishi, along with some edible seaweeds, contain significant amounts of beta-glucans. Dietary fibers, including soluble fibers, found in oats, legumes, and certain fruits, dissolve in water to form viscous gels, which can modulate blood glucose levels and lower serum cholesterol [14,15,16]. Insoluble fibers, present in whole grains and vegetables, promote bowel regularity and prevent constipation. Prebiotic polysaccharides, such as inulin and fructooligosaccharides, selectively stimulate the growth of beneficial gut microbiota, enhancing gut health and systemic immunity. Dietary fibers have little to no nutritional value, but modulate the transit of food in the gastrointestinal tract [5].

The cardiovascular benefits of polysaccharides are multifaceted. Soluble fibers, by their high viscosity and water-binding properties, are known to bind bile acids in the intestine, facilitating their excretion and thereby reducing circulating cholesterol levels [17]. This mechanism is pivotal in lowering the risk of atherosclerosis, a major CVD contributor. Moreover, the fermentation of dietary fibers by gut microbiota produces short-chain fatty acids (SCFAs), such as butyrate, propionate, and acetate, which have anti-inflammatory properties and improve endothelial function. These SCFAs also play a role in regulating lipid metabolism and glucose homeostasis, further mitigating CVD risk. Interestingly, a popular low-carbohydrate diet also reduces the dietary intake of polysaccharides, which was not considered to lower cardiovascular risk in two-year observations of randomized clinical trials [18].

A group of potential sources of polysaccharides is marine macroalgae, also known as seaweed. They are traditionally harvested in Asian countries and used in the food industry, agriculture, and medicine [19]. The key effects that can be linked to CV risk are generally characterized as antioxidant activity, microbial modulation, and obesity prevention by lowering food intake.

However, not all polysaccharides confer health benefits. Refined starches, prevalent in processed foods, can lead to rapid spikes in blood glucose and insulin levels, contributing to insulin resistance, obesity, and metabolic syndrome [20]. These conditions are significant CVD risk factors [5]. Additionally, excessive consumption of certain soluble fibers through supplements can interfere with absorbing essential minerals, potentially leading to deficiencies and gastrointestinal discomfort. Others can have ambiguous effects on the microbiome, resulting in decreased diversity, not necessarily promoting beneficial bacteria.

This review aims to explore the dualistic nature of polysaccharides in health, emphasizing their beneficial roles in reducing CVD risk factors and highlighting the potential adverse effects of specific types (as presented in Figure 2). Their impact on cardiovascular health, supported by epidemiological evidence, leads to an understanding of particular mechanisms, which translates to potential dietary interventions that can be subjected to optimization. For facilitated understanding of polysaccharide group activity, the summarized information on their influence is presented in Table 1.

## 2. Dyslipidemia

Low-density lipoprotein cholesterol (LDL) is considered a major risk factor for atherosclerotic CVD [29]. Brown et al. performed a metanalysis on soluble fibers, indicating that the consumption of 2–10 g/day led to a statistically significant lowering of total cholesterol (TC) and LDL [23]. The effect was not observed for high-density lipoprotein cholesterol (HDL) and triglycerides (TG). The differences between sources of soluble fiber were not statistically important, also considering background fat content. The findings complied with recommendations by Reynolds et al. indicating that intake of 15–30 g fiber/day improves lipid profile in diabetic patients, especially in terns if LDL, TC, and TG [22]. The 2019 ESC/EAS guidelines for the management of dyslipidemia recommend a dietary fiber intake of 25–40 g/day, including ≥7–13 g of soluble fiber, preferably from wholegrain products, e.g., oats and barley [4]. Comparison of polysaccharide effects in lipid profiles is briefly presented in Table 2.

The studies performed by the group reported by Fernandez et al. on guinea pigs revealed that pectin, guar gum, and psyllium-fed guinea pigs were observed to have faster catabolism and cholesterol clearance, resulting in increased LDL-ApoB 100 [29]. The meta-analysis performed by Li et al. of 17 randomized clinical trials on a homogenic group of dyslipidemic patients resulted in no difference in reducing major cardiovascular events with oat-based product supplementation in the diet [30]. Lipid profiles were altered by lowering LDL and TC, and little through the lowering of HDL and TG, resulting in the lowering of coronary heart disease incidence by 4%. However, direct evidence is lacking, and the observation was classified as moderate in clinical meaning.

A meta-analysis focused on the effects of soluble fiber supplementation on blood lipid parameters in adults, which included 181 RCTS and 14,505 participants, showed in the overall analysis that there was a significant reduction in LDL, TC, TG, and apolipoprotein B (apoB) after soluble fiber supplementation. The dose-response analysis indicated that each 5 g/day increase in soluble fiber supplementation had a significant reduction effect on TC and LDL [31].

Oats are one of the main everyday sources of dietary fiber. A meta-analysis of 17 trials with 1731 subjects showed that oat intervention (based on both beta-glucan-based and bran-based and wholegrain oat products) compared to the placebo or usual diet resulted in a significant reduction in LDL and TC, with little effect on HDL and TG [30]. On the other hand, one study included in the aforementioned meta-analysis indicated that oat-based products made no significant difference in major cardiovascular events. As already indicated, oats are a good source of beta-glucan. A meta-analysis of 58 trials with total of 3974 participants showed that a median dose of 3.5 g/day of oat beta-glucan significantly lowered LDL, non-HDL-cholesterol, and apoB. The 3.5 g/day intake lowered the abovementioned cholesterol fractions by −0.19 mmol/L and −0.2 mmol/L, respectively, which equates to 4.2%. Increasing the dosage to 6.9 g/day of barley beta-glucan lowered LDL and non-HDL-cholesterol by 7% compared with control diets. As intake in modern society is very often supplemented by alternative sources, high-viscosity fibers (glucomannan, psyllium) were also reported to lower LDL and non–HDL-cholesterol with dosages of 10 g/day and 3–4 g/day, respectively [32].

A meta-analysis focused on beta-glucan consumption in hypercholesterolemic patients, which included 17 RCTs covering 916 subjects, confirmed that beta-glucan consumption significantly lowered TC and LDL concentration, while there were no significant differences in HDL, TG, and glucose [33]. Further investigation of 28 studies, totaling 1494 subjects, confirmed that whole oats intervention decreased TC and LDL [34]. This meta-analysis also showed that isolated beta-glucan interventions decreased TC and TG; however, HDL was not altered by either oat or isolated β-glucan [34]. Barley, as another source of beta-glucan, in a meta-analysis of 8 trials covering 391 patients, showed that it significantly lowered TC, LDL, and TG, but did not alter HDL significantly [35]. A meta-analysis of 11 RCTs confirmed that barley and beta-glucan isolated from barley lowered TC and LDL concentrations. However, there were no significant subgroup differences by the food matrix [36]. On the other hand, a similar interesting approach was presented in a meta-analysis which focused on the different delivery matrices of beta-glucan. It confirmed that consuming ≥3 g/day of beta-glucan for at least 3 weeks significantly reduced TC and LDL in mildly hypercholesterolemic individuals, while no significant difference was observed in TG and HDL. It indicated that the effects of food matrices with both solid and liquid products where beta-glucan was incorporated were ranked as the best way to exert its beneficial properties, compared to only liquid or solid products alone [37]. These differences might be caused by the fact the first mentioned meta-analysis was focused on barley-derived beta-glucan, while the second one did not narrow down the source of beta-glucan.

Chia seeds are another source of dietary fiber. A meta-analysis of 10 clinical trials showed that chia consumption decreased TC, TG, and LDL, and increased HDL; however, the results were not statistically significant [38]. For psyllium, a meta-analysis of 28 trials covering a total of 1924 participants showed that supplementation of 10.2 g psyllium significantly reduced LDL, non-HDL, and apoB [39]. Psyllium can be an addition to other cholesterol-lowering actions, and adding psyllium fiber resulted in reductions in LDL equivalent to doubling the statin dose [40]. A meta-analysis of 21 studies, covering 1030 subjects with mild-to-moderate hypercholesterolemia, confirmed that consumption of psyllium lowered serum TC and LDL. What is more, a significant dose–response relationship was found between doses from 3 to 20.4 g/day [41]. Moreover, a meta-analysis covering 384 mild-to-moderate hypercholesterolemia patients based on eight studies showed that 10.2 g psyllium/day added to a low-fat diet lowered serum TC by 4%, LDL by 7%, and the ratio of apoB to apo A-I by 6% compared to placebo, i.e., a low-fat diet alone. No significant differences were observed for HDL or TG levels [42]. Olson et al., in a meta-analysis of 12 studies, observed that subjects who consumed a psyllium-enriched cereal had lower TC and LDL concentrations than the ones who ate a control cereal. HDL levels were not significantly changed [43]. Additionally, data sourced from 10 trials covering 268 participants showed that a diet rich in non-soy legumes, which are sources of fiber, reduced TC and LDL [44]. Brown seaweeds, as a source of marine polysaccharides (mostly laminarines and sulfonated ones), were evaluated by Shin et al. in a meta-analysis, which concluded that fucoidans do not have a significant effect on dyslipidemia markers or obesity based on seaweed supplementation in diet [45].

Familial hypercholesterolemia is a genetic condition resulting in significantly elevated LDL levels and high cardiovascular risk without a previous lifestyle background. The Cochrane Database analysis, focused on dietary interventions in familiar hypercholesterolemia, underlined that in the available literature, only short-term, secondary outcomes could be assessed. However, it indicated that guar gum, when given as an add-on therapy to bezafibrate, reduced TC and LDL levels as compared to bezafibrate alone [46]. The inulin-type fructans-related meta-analysis of 20 RCTs with 607 adults showed that the supplementation reduced LDL [27]. A meta-analysis of 14 trials encompassing 1108 participants showed that chitosan supplementation significantly improved TC and LDL concentrations, while no significant changes were seen in HDL and TG [47].

In summary, soluble fiber supplementation plays a significant role in improving lipid profiles, particularly by reducing TC and LDL, which are major risk factors for CVD in various populations. The effects on HDL and TG are often minimal or not statistically significant regardless of the sources (oats, barley, psyllium) and populations. The beneficial effects of soluble fiber appear dose-dependent, with higher intakes yielding greater reductions in LDL and TC. Psyllium, in particular, shows potential as an adjunct therapy to standard cholesterol-lowering treatments, such as statins. Despite the overall positive trends observed, further research to clarify the impact of polysaccharides on dyslipidemia is needed. Nevertheless, it can be concluded that fiber in particular is a valuable dietary intervention for risk management.

## 3. Hypertension

One of the most worth mentioning and widespread recommendations in hypertension management is the introduction of Mediterranean and/or Dietary Approaches to Stop Hypertension (DASH) dietary patterns in cardiovascular disease prevention. These have been reported to reduce cardiovascular risk [25]. The DASH pattern is high in carbohydrates (including fiber), moderate in protein, and low in total fat and saturated fatty acids, and are also high in potassium, calcium, and magnesium, and low in sodium, with a high intake of polysaccharide-rich fruits, vegetables, and whole grains [25]. High-fiber diets like DASH are often rich in potassium, which can help lower blood pressure by counteracting the effects of sodium and promoting vasodilation.

A meta-analysis performed by Ghavami et al. on 83 trials with 5985 participants evaluated the supplementation of soluble fiber on systolic blood pressure by a significant reduction in systolic blood pressure (SBP) and diastolic blood pressure (DBP). SBP and DBP had a larger reduction in the guar gum and inulin subsets [48]. The effect was observed in dose-responsive manners in hypertensive participants, as SBP decreased proportionally with the increase in soluble fiber in the course of supplementation up to 20 g/day. With moderate certainty, high fiber intake reduces mortality (RR = 0.75, (95% confidence interval, CI 0.58–0.97)), irrespective of cardioprotective therapies [49]. Another meta-analysis, which focused particularly on psyllium impact and covered 11 trials with 592 participants, showed a significant reduction in SBP after psyllium supplementation [50]. Moreover, this relationship was stronger in participants with higher baseline blood pressure. The positive impact of psyllium supplementation on SBP was confirmed in another meta-analysis [51]. Flaxseed is another popular source of fiber. A meta-analysis of 14 trials by Khalesi et al. indicated that flaxseed supplementation significantly reduced SBP and DBP [52]. Interestingly, these results were not influenced by higher baseline blood pressure. For oat consumption, a meta-analysis of 21 RCTs involving 1569 participants proved that consuming oats significantly reduced SBP, particularly in hypertensive participants [53]. There was no significant impact noted for DBP in the overall group, but subgroup analysis revealed that this relationship was significant in participants who were prehypertensive at baseline. These effects were observed when the consumption was at a level of ≥5 g/day β-glucan, or over ≥8 weeks [53]. A similar pattern has been observed for chitosan, as in a meta-analysis of eight trials with 617 total participants [54]. Although overall chitosan supplementation did not significantly change SBP or DBP, the subgroup analysis indicated that chitosan consumption significantly reduced DBP in higher doses (>2.4 g/day) or, interestingly, shorter-term (<12 weeks) arms [54].

Besides the mentioned binding of fiber and glucans to bile acids and excretion of cholesterol, dietary fibers can improve endothelial function and increase the production of nitric oxide, a vasodilator that helps in maintaining blood vessel elasticity and lowering blood pressure, as well as reducing inflammation and oxidative stress [55]. Low fiber with high fat in a cafeteria diet, which accurately reflects a Western obese diet, results also in endothelium dysfunction, with a significant reduction of vasodilatory response to acetylcholine [55].

Similarly to dyslipidemia, fiber supplementation has been shown to be beneficial and to consistently bring improvement in hypertensive individuals. Meta-analyses have highlighted that sources like oats, barley, psyllium, and beta-glucans are effective across various populations and dosages, as summarized in Table 3. Meta-analyses have provided information showing that soluble fiber supplementation significantly reduces both SBP and DBP, with larger reductions observed for guar gum, inulin, and psyllium. The dose-dependent effects suggest that higher intakes of fiber, up to 20 g/day, result in greater reductions in blood pressure. Additionally, dietary fiber improves endothelial function and enhances nitric oxide production, which supports vasodilation. High fiber intake lowers hypertension and also improves cardiovascular outcomes regardless of other cardioprotective therapies.

## 4. Obesity

A high intake of refined starches (such as white bread or white rice) has been associated with poor cardiometabolic health and an increased risk of CVD events [25]. It is crucial to avoid them, especially in carbohydrate-restricted diets, as their intake increases the risk of cardiovascular events [56,57]. This is also an important factor in calorie-focused diets, where fat limitation can result in a reduction of healthful vegetable oils and replacement with starch. Generally, dietary fiber increases the feeling of fullness (satiety) by slowing gastric emptying and prolonging digestion. This reduces overall caloric intake, contributing to weight loss or prevention of weight gain, both of which are critical factors in reducing CV risk, which was concluded in Table 4.

Inulin was also considered in trials comparing dietary interventions to a control diet, indicating a TG decrease, but meta-analyses did not indicate statistical significance [27]. A similar situation was indicated for TC, except for trials performed in type 2 diabetes mellitus (T2DM) patients. For obese groups of patients, inulin was also reported in three trials to lower LDL and improve lipid profile. It also acts by stimulating the feeling of being full, as a low-calorie dietary fiber. Generally, CVD risk is lowered also by the influence of fibers, leading to weight reduction, which follows their influence on hypertension, lowering LDL cholesterol and inflammation via colonic microbiota [58,59]. Beneficial gut bacteria produce SCFA such as butyrate, which have anti-inflammatory effects that can reduce systemic inflammation and consequently lower cardiovascular risk [60]. These SCFAs improve endothelial function and increase nitric oxide production, a vasodilator that helps maintain blood vessel elasticity and lower blood pressure. A similar mode of action has been assigned to pectins, which are abundant cell wall polysaccharides in plants [61].

Chitosan’s role in overweight patients was evaluated in 15 trials on 1219 participants by Jull et al., and it was concluded that chitosan may have a small effect on body weight, but this effect was considered to be minimal. Chitosan is a natural polymer in the exoskeletons of crustaceans and has several properties in the biomedical and food industries. Its effects on weight loss in overweight adults were studied in a metanalysis of 111 trials [62]. The mean difference was −1.70 kg, alongside −3.7 and 1.36 kg for psyllium and glucomannan, respectively, compared with the placebo [62]. The effect, however, was observed in trials longer than 12 weeks, indicating chronic intake is crucial to acquire the desired dietary effect [54]. In such prolonged use, other chitosan properties need to be considered; it is liable to alter the balance of intestinal microbiota, as it has antimicrobial activity by binding to cell membranes, interfering with enzymes and metal chelation [63]. Antimicrobial activity can be more selective towards pathogenic bacteria while sparing beneficial gut bacteria, but further studies are still needed in this area, because prolonged use may disrupt the balance of gut microbiota, potentially leading to reduced diversity and dysbiosis.

Regarding seaweed polysaccharides and oligosaccharides, sulfated polysaccharides extracted from *Undaria pinnatifida* (Phaeophyceae) can modulate the gut microbiota, thereby inhibiting weight gain and lipid metabolism, which could benefit weight loss, as indicated by Zhang et al. [64]. Similar observations, alongside anti-inflammatory action related with fucoidins and alginates, often co-exist in sources rich in polyphenols [65]. Obesity is a main factor in metabolic syndrome, and the proper management of inflammation is crucial. Alginate oligosaccharides have been proven to attenuate the postprandial blood glucose response by the consumption of alginate glucose drinks and a reduction in cholesterol and glucose absorption from the intestines. Animal tests have also provided information on improving insulin sensitivity, but further trials are needed [66].

Algal polysaccharides were also considered in mitigating atherosclerosis. Fucoidan has anti-inflammatory and anticoagulant effects, modulating a decrease in LDL-C and TG [67]. Laminarin contributes by reducing oxidative stress and inflammation, key factors in atherosclerosis development, and both can help regulate lipid metabolism, aid in weight management, and promote gut health, which indirectly lowers cardiovascular risk [67].

In obesity, poor cardiometabolic health has been linked to refined starches, which should be avoided in a calorie-focused diet and replaced with beneficial fats. Dietary fibers, in general, lower CVD risk by promoting weight loss, improving blood pressure, reducing LDL levels, and modulating the gut microbiota to reduce inflammation, similar to pectins and chitosan.

## 5. Diabetes

Diabetic patients are another group of interest in terms of cardiovascular risk, which is possible to modify multi-directionally, as presented in Table 5. It is worth noting that diabetes itself raises CVD risk to high or extremely high [1]. In T2DM, a meta-analysis of 11 papers demonstrated a significant reduction in total cholesterol and low-density lipoprotein cholesterol following guar gum supplementation. Moreover, the subgroup analysis showed that ≥20 g/day of guar gum led to a notable decrease in triglyceride levels versus < 20 g/day. Guar gum supplementation had no effects on high-density lipoprotein cholesterol [68]. The other way round assessment, i.e., glycemic control in hypercholesterolemic individuals, based on 12 trials with a total of 603 subjects, indicated that beta-glucan consumption did not significantly affect measures of glycemic control [69]. Studies have shown also that supplementation with inulin can improve insulin sensitivity in overweight adults by increasing the abundance of *Bacteroides* and promoting a bifidogenic effect [70].

Observational studies have shown that a higher intake of dietary fiber is associated with a reduced risk of CVD as well as T2DM and obesity [5]. Diabetes mellitus is a group of metabolic disorders of carbohydrate metabolism, resulting in chronic hyperglycemia [71]. Insulin resistance resulting from chronic inflammation accompanying obesity leads to relative insulin deficiency, which forces the pancreatic islets to increase insulin secretion. In the longer term, it can lead to beta islet degradation. Progressive loss of beta cells and their function manifests clinically as hyperglycemia, particularly connecting polysaccharides with positive and negative effects. Without proper management of diabetes, chronic complication risk rises, possibly leading to microvascular (retinopathy, nephropathy, and neuropathy) and macrovascular complications (coronary artery disease, atherosclerosis, peripheral artery disease, etc.), although rates of progression may differ. The current recommendations for diabetes-related carbohydrate management are based on proper counting, which is an evidence-based approach based on World Health Organization (WHO) guidelines [72,73,74]. Low-carbohydrate diets in diabetes have been shown to reduce the risk of cardiovascular diseases, hypertension, and insulin resistance. However, given their restrictive nature, it is considered that they can also lower the intake of fiber-rich elements, causing an imbalance in macronutrient intake, suboptimal micronutrient intake, and increased disease risk over time [18].

The antidiabetic effect of macroalgae was postulated by Tavares et al. in their work on its possible applications in healthcare [75]. Sulfate polysaccharides, such as agar and carrageenan, were reported to inhibit digestive enzymes (namely, α-amylase and α-glucosidase), resulting in lowering glycemia. Also, as mentioned in the Obesity section, alginates and fucoidins have the potential to lower inflammation. Algae are also a rich source of fibers, facilitating satiety, as indicated earlier.

Beta-glucans, composed of D-glucose units connected with beta-glycosidic bonds, are reported to have a beneficial impact on the gastrointestinal and immune systems [20]. Such nonspecific immune system responses and immunostimulation have been reported in humans and animals [76,77]. They are not digested after ingestion, and their enzymatic breakdown is slower than starch, which results in reduced glycemic spikes, which is desired in proper diabetes management [18]. Oat beta-glucans present a linear dose-response impact on acute glucose and insulin response after carbohydrate meals [78]. Moreover, a meta-analysis of 18 studies confirmed that oat intake resulted in a greater decrease in fasting glucose insulin and hemoglobin A1c compared to the control [79]. Similar conclusions were made in the systematic review by Andrade et al., which indicated that doses of beta-glucans of 6 g/day for >4 weeks led to improvements in blood glucose levels. However, blood glucose levels do not reach normal levels with the application of beta-glucans alone [80].

Starches are digested rapidly (high glycemic index) in the mouth and stomach, and this is particularly concerning for individuals with diabetes. However, the long-term effects of starchy vegetables have been evaluated, and have been shown to have uncertain cardiometabolic effects [81]. Some starches, known as resistant starches, resist digestion in the small intestine and reach the large intestine to become fermented by microorganisms. One of the most common forms of resistant starches in use is high amylose corn-starch (HAM-RS2), which lowers caloric intake by replacement of flour carbohydrates. It has been reported to improve insulin sensitivity among individuals at risk of or with type 2 diabetes, thus lowering cardiovascular risk, with a high level of confidence in 2023 ESC guidelines for the management of cardiovascular disease in patients with diabetes [26]. By reducing insulin resistance, fiber intake decreases the risk of developing type 2 diabetes and associated cardiovascular conditions. However, while carbohydrates lower overall CVD risk, the rapidly digestible starch and sugar provided by whole vegetables and fruit may need to be taken into consideration in people with diabetes if intakes are very high [74]. Additionally, an interesting remark on the role of polysaccharides was postulated by Castro-Acosta et al., as their presence (e.g., starch) enables polyphenols to bind to them instead of the digestive enzymes, diminishing anthocyanins’ effects of lowering glycemia [20].

What is more, in a meta-analysis of 10 trials including 1473 participants, chitosan supplementation led to a significant reduction in fasting glucose levels and hemoglobin A1c levels, with no effect on insulin levels [82]. Further analysis showed significant reductions in fasting glucose levels after supplementation of 1.6–3 g/day or for >13 weeks [82].

Diabetic patients are liable to an elevated risk of CVD and can particularly benefit from the use of dietary fiber. Polysaccharide supplementation in T2DM has shown a reduction in TC and LDL, while also decreasing TG. On the other hand, beta-glucan consumption, while not significantly impactful with regard to glycemic control, has been linked to improved gastrointestinal and immune functions, as well as reduced postprandial glycemic spikes in diabetes management. While low-carbohydrate diets can reduce CVD risk and improve insulin sensitivity, they may also reduce fiber intake, which is crucial in preventing long-term complications. Resistant starches have demonstrated the ability to improve insulin sensitivity, thus lowering CVD risk. Additionally, polysaccharides can impact the bioavailability of polyphenols, potentially diminishing their glycemic-lowering effects.

## 6. Conclusions

In terms of selected CVD risk factors, polysaccharides such as dietary fiber and beta-glucans have a positive impact on lipid profiles, particularly on TC and LDL levels, which are mediated by lipid absorption and microbiome modulation. A positive impact on blood pressure control, mainly SBP, has also been noted. What is more, polysaccharides present beneficial properties for body mass control, which is important in the context of obesity prevention, and in this way, they indirectly contribute to the control of other risk factors secondary to it, alongside chitosan or inulin dietary addition. For diabetes, the reported impact depends on the polysaccharide type, as refined starches can lead to rapid glucose spikes, which are not desired in T2DM patients, and prolonged changes in ingested carbohydrates need to be evaluated in terms of their quality or overuse. On the other hand, dietary fiber and beta-glucans can stabilize acute glucose blood levels as well as fasting glucose and hemoglobin A1c. To sum up, fiber and beta-glucan intake is considered an advised approach to CVD prevention.

## Figures and Tables

**Figure 1 polymers-17-00405-f001:**
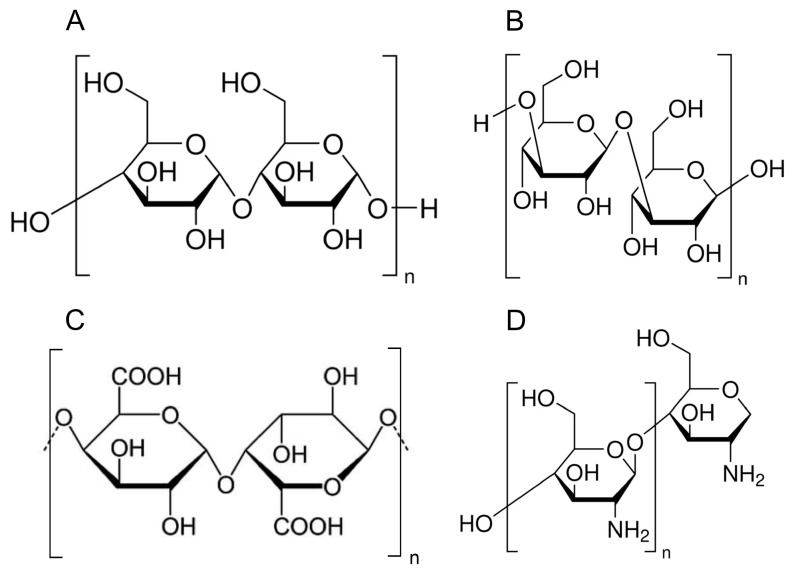
Examples of structural formulae of described dietary polysaccharides: (**A**) starch, (**B**) beta-D-glucan, (**C**) pectin, (**D**) chitosan.

**Figure 2 polymers-17-00405-f002:**
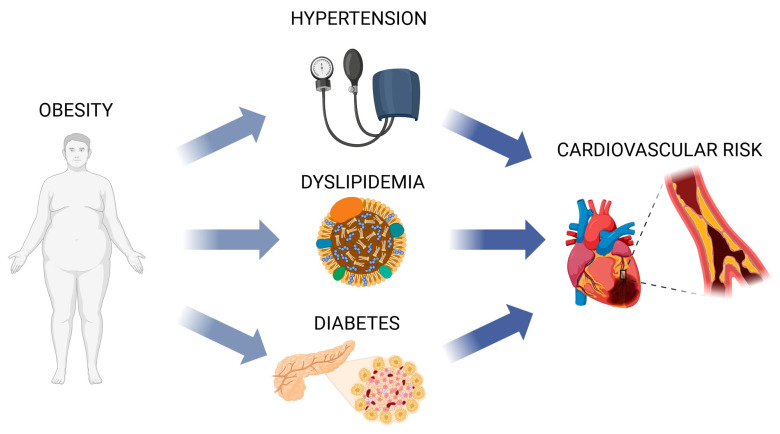
The main dietary-manageable disorders related to increased CVD risk. Created with Biorender.com.

**Table 1 polymers-17-00405-t001:** Consumption of polysaccharides and their influence on cardiovascular risk constituents.

Polysaccharides	Sources	Recommendation	Result
Soluble Fiber	Oats, barley, beans, lentils, peas, apples, citrus fruits, carrots, psyllium husk	intake of 4–10 g/day [21,22,23]	protective in cardiometabolic complications decrease of mortality lowering glucose lowering TC, LDL, TG
Insoluble fiber	Whole grains (wheat bran, brown rice), nuts, seeds, vegetables (cauliflower, green beans, potatoes)	70–75% of total fiber [21,22,23]	
Beta-glucans	Oats, barley, mushrooms (e.g., shiitake, maitake), yeast	3–6 g/day [24]	lowers LDL and non-HDL
Refined starches	White bread, white rice, pastries, many processed foods, crackers, cakes, and some breakfast cereals	should be minimized 1–2 servings/day [25]	worsening of cardiometabolic health and an increased risk of CVD events
Starch	Potatoes, corn, peas, pasta, bread, and other grains	No recommendation	high glycemic index, glycemia spikes
Resistant starch	Green bananas, legumes, cooked and cooled potatoes, rice, pasta, high-amylose cornstarch	15–60 g/day [26]	decrease of available energy lower blood glucose increase of TG
Inulin	Chicory root, onions, garlic, leeks, asparagus, artichokes, bananas	5–10 g/day [27]	lowering TC and LDL Modulation of inflammation enhance the growth of beneficial gut bacteria
Chitosan	Exoskeleton of crustaceans (crab, shrimp, lobster)	no recommendation, considered 1–6 g/day [28]	Decreasing body weight Has prebiotic and antimicrobial properties

**Table 2 polymers-17-00405-t002:** Summary of dietary components and their effects in dyslipidemic patients.

Dietary Component	Key Effect in Dyslipidemia
Soluble fibers	↓TC, ↓LDL
Beta-glucans	↓LDL, ↓non-HDL, ↓apoB
Guar gum	↓TC, ↓LDL (with therapy of familial hypercholesterolemia)
Inulin-type fructans	↓LDL
Chitosan	↓TC, ↓LDL
Psyllium	↓LDL, ↓non-HDL, ↓apoB

↓—lowering effect, ↑—increasing effect.

**Table 3 polymers-17-00405-t003:** Summary of dietary components and their effects in hypertensive patients.

Dietary Component	Key Effect in Hypertension
Soluble fibers	↓SBP, ↓DBP, improves endothelial function, ↓Inflammation
Beta-glucans	↓SBP
Chitosan	↓DBP (in high doses)
Psyllium	↓SBP

↓—lowering effect, ↑—increasing effect.

**Table 4 polymers-17-00405-t004:** Summary of dietary components and their effects in obese patients.

Dietary Component	Key Effects in Obesity
Dietary fiber	↑satiety, ↓gastric emptying, ↓caloric intake, ↑body weight loss, ↓CV risk, ↓LDL-C, ↓inflammation
Inulin	↓LDL-C, improves lipid profile
Pectins	improve endothelial function, ↓inflammation, ↓CV risk
Chitosan	↓body weight in overweight patients, can disrupt gut microbiota balance in prolonged use
Psyllium and Glucomannan	↓body weight
Alginates	↓cholesterol and glucose absorption

↓—lowering effect, ↑—increasing effect.

**Table 5 polymers-17-00405-t005:** Summary of dietary components and their effects in diabetic patients.

Dietary Component	Key Effects in Diabetes
Low carbohydrate diets (High fiber)	↓CVD risk, ↓hypertension, ↓insulin resistance
Beta-glucans	↓postprandial glycemic spikes, ↓fasting glucose, ↓insulin, ↓HbA1c
Chitosan	↓fasting glucose, ↓HbA1c
Resistant starches	↑insulin sensitivity, ↓caloric intake, ↓CVD risk
Guar gum	↓TC, ↓LDL-C, ↓TG

↓—lowering effect, ↑—increasing effect.

## Data Availability

Data sharing is not applicable. Data are contained within the article and derived from publicly published papers.

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
