# Peer review of "Polysaccharides: The Sweet and Bitter Impacts on Cardiovascular Risk"

_polymers, 2025, doi:10.3390/polym17030405_

Round 1

Reviewer 1 Report

Comments and Suggestions for Authors  

Kalisz et al. discuss in this review article the use of different starch products to prevent heart disease. The article is engaging and suitable for publication in this journal, provided the following revisions are made:

i. Each section should include a conclusive statement.

ii. The review should explain the scientific mechanisms by which consuming various fiber-containing carbohydrates reduces heart disease risk.

iii. Sections 4 and 5 need to be expanded with more detailed information.

Author Response

Kalisz et al. discuss in this review article the use of different starch products to prevent heart disease. The article is engaging and suitable for publication in this journal, provided the following revisions are made:

Dear Reviewer,

Thank you for your time and effort in revising our manuscript. We appreciate the comments listed, and we answered them point-by-point below.

  1. Each section should include a conclusive statement.

Conclusive statements for each section were added to the manuscript

  1. The review should explain the scientific mechanisms by which consuming various fiber-containing carbohydrates reduces heart disease risk.

The description of possible mechanisms of polysaccharides on CV risk was provided in lines 82-100, and also lines 120-124, 341-346, 396-399. However, mechanisms of action were expanded in lines 38-42, 276-279, 224-226, 291-293

“Elevated blood pressure and diabetes promote endothelium dysfunction and damage, which along with elevated low-density cholesterol (LDL) levels leads to atherosclerosis development. If atherosclerotic lesions are present in coronary arteries, such a condition is defined as coronary heart disease. “

“Generally, dietary fiber increases the feeling of fullness (satiety) by slowing gastric emptying and prolonging digestion. This reduces overall caloric intake, contributing to weight loss or prevention of weight gain, both of which are critical factors in reducing CV risk. “

“High-fiber diets like DASH are often rich in potassium, which can help lower blood pressure by counteracting the effects of sodium and promoting vasodilation.”

“These SCFAs improve endothelial function and increase nitric oxide production, a vasodilator that helps in maintaining blood vessel elasticity and lowering blood pressure.”

iii. Sections 4 and 5 need to be expanded with more detailed information

More information on other polysaccharide sources (especially seaweed) was added to the sections 2-5.

Reviewer 2 Report

Comments and Suggestions for Authors

1. The authors need to rewrite the manuscript with more critical analysis of the results included in this review.

2. Structures in figure 1 can be drawn appropriately with correct specifications.

3. Redrawing the figures, including more figures and tables to explain the data may help to improve the quality of the paper.

4. Reference list can be formatted according to the publication standard.

5. Reference list is long. Four and half pages reference list for a review of effectively thirteen pages in total. Irrelevant references can be removed.
There are few grammatical errors that can be fixed to improve the quality of the draft such as:

1. Repeated sentences. E.g. the first sentence in the abstract and the first sentence in an introduction is almost same.

2. Inconsistent use of units. E.g. at some places the unit used is g/day and some places it is g/d (line 133, line 164)

3. Most of the places the sentences started with same kind of wording: E.g. the word “moreover” was used frequently to start the sentence. (line 181-189)

4. There is a random “N” in the figure 1

5. The data presentation is poor. The paragraphs are very small randomly: E.g. lines 148-158 and lines 187-204.

Comments on the Quality of English Language

There is a need to improve the quality of English in this draft. please see the comments to the authors

Author Response

  1. The authors need to rewrite the manuscript with more critical analysis of the results included in this review.

Dear Reviewer,

Thank you for your insightful feedback and effort in reviewing our manuscript. We added conclusive statements for each section of the manuscript to highlight the most important results in the lines: 214-223, 269-279, 340-344 and 421-431.

  1. Structures in figure 1 can be drawn appropriately with correct specifications.

Figure 1 was completely replaced with new one.

  1. Redrawing the figures, including more figures and tables to explain the data may help to improve the quality of the paper.

Figure 1 was replaced with a new version, and the manuscript was extended with tables for each section, summarizing briefly the information from the text in lines 212, 267, 338, 384.

  1. Reference list can be formatted according to the publication standard.

The reference list was changed automatically by Zotero which was overlooked by us and we apologize for the inconvenience. The correction to maintain the Polysaccharides (MDPI) standard was made.

  1. Reference list is long. Four and half pages reference list for a review of effectively thirteen pages in total. Irrelevant references can be removed.

We believe the references used were relevant to the paper, and removing them can influence the quality of the information presented. Unfortunately, we also have the opposite commentary of the other reviewer, and we cannot comply with both demands.

There are few grammatical errors that can be fixed to improve the quality of the draft such as:

  1. Repeated sentences. E.g. the first sentence in the abstract and the first sentence in an introduction is almost same.

The abstract and introduction were rephrased to avoid repeated sentences. (e.g. line 11-12, 29-30)

2. Inconsistent use of units. E.g. at some places the unit used is g/day and some places it is g/d (line 133, line 164)

The units were revised in the whole manuscript and unified.

  1. Most of the places the sentences started with same kind of wording: E.g. the word “moreover” was used frequently to start the sentence. (line 181-189)

Frequently sentence starters like “moreover”, “however”, “although” etc. were corrected in the manuscript to improve readability.

  1. There is a random “N” in the figure 1

Figure 1 was completely replaced with a new one.

  1. The data presentation is poor. The paragraphs are very small randomly: E.g. lines 148-158 and lines 187-204.

The sections included small paragraphs with poor presentation were rephrased. Additionally, tables were added to each section to provide a better understanding in lines 212, 267, 338, 384

Reviewer 3 Report

Comments and Suggestions for Authors

General comments

Cardiovascular risk, connected with the probability of developing cardiovascular diseases such as coronary artery disease or stroke, is shaped by both non-modifiable and modifiable factors, including dietary patterns. Polysaccharides are essential components of dietary nutrition because they play a critical role in energy consumption and other physiological functions. This review examines the effects of dietary polysaccharides on cardiovascular health depending on their type and source. Of course, such studies are important and relevant.

However, the proposed review should be strengthened both in content and in the number of works analyzed. For example, the review (reference [5] on a related topic, Trautwein EA, McKay S. The Role of Specific Components of a Plant-Based Diet in Management of Dyslipidemia and the Impact on Cardiovascular Risk. Nutrients 2020;12:2671. https://doi.org/10.3390/nu12092671) is more detailed and contains 3 tables, 100 references.

In addition, to improve the article, it may be recommended:

1.                  The reviewed version of the manuscript should be made more visual. For example, the text given on page 2 can be presented as a color scheme.

2.                  The number of polysaccharides considered should be increased, for example by using polysaccharides obtained from seaweed.

3.                  Since the article is related to nutrition, a table could be added in which the polysaccharides in question should be linked to their sources, and how they are used for nutritional and medicinal purposes could be considered.

4.                  Table 1 should be supplemented with references to literary sources.

5.                  It would also be convenient to summarize the recommendations offered by various authors by compiling another table.

Author Response

Cardiovascular risk, connected with the probability of developing cardiovascular diseases such as coronary artery disease or stroke, is shaped by both non-modifiable and modifiable factors, including dietary patterns. Polysaccharides are essential components of dietary nutrition because they play a critical role in energy consumption and other physiological functions. This review examines the effects of dietary polysaccharides on cardiovascular health depending on their type and source. Of course, such studies are important and relevant.

However, the proposed review should be strengthened both in content and in the number of works analyzed. For example, the review (reference [5] on a related topic, Trautwein EA, McKay S. The Role of Specific Components of a Plant-Based Diet in Management of Dyslipidemia and the Impact on Cardiovascular Risk. Nutrients 2020;12:2671. https://doi.org/10.3390/nu12092671) is more detailed and contains 3 tables, 100 references.

Dear Reviewer,

Thank you for your time and effort in reviewing our manuscript. The mentioned paper by Trautwein et al. considers not only polysaccharides, but also dietary fats, phytosterols etc. Due to the fact that we focused only on polysaccharides, as it is a scope of the journal, the number of analyzed works is in fact smaller. We improved the readability by presenting data in tables, which we hope improves overall quality of the manuscript. Our detailed answers are in point-by-point manner below.

In addition, to improve the article, it may be recommended:

  1. The reviewed version of the manuscript should be made more visual. For example, the text given on page 2 can be presented as a color scheme.

We improved the readability by presenting data in tables 1-4, summarizing information of each section to provide a better understanding in lines 212, 267, 338, 384

  1. The number of polysaccharides considered should be increased, for example by using polysaccharides obtained from seaweed.

The health-related works were evaluated, as a supplement to the diet and particular information on polysaccharides were added to the manuscript in lines 96-100, 159-166, 198-201, 322-337 and 378-383.

  1. Since the article is related to nutrition, a table could be added in which the polysaccharides in question should be linked to their sources, and how they are used for nutritional and medicinal purposes could be considered.

Sources of each polysaccharide were added to the summarizing table 5 (line 432)

  1. Table 1 should be supplemented with references to literary sources.

References were added to the Table 1 (now table 5)

  1. It would also be convenient to summarize the recommendations offered by various authors by compiling another table.

Additional tables referring to the most important action were added to each paragraph of the manuscript g in lines 212, 267, 338, and 384, beside the recommendations in Table 5.

Round 2

Reviewer 1 Report

Comments and Suggestions for Authors

I am delighted with the authors' revision. This manuscript should be accepted for publication in its present form.

Reviewer 2 Report

Comments and Suggestions for Authors

The authors have done a good job of revising the manuscript. The figures are good and the additional tables simplify understanding of the data. The detailed explanation gives more soundness about the scientific data.

Reviewer 3 Report

Comments and Suggestions for Authors

In general, the comments made to the authors have been taken into account. The article has become more illustrative. The number of references has increased.

 I believe that the article can be published in its current form.
